# miR-208b Reduces the Expression of Kcnj5 in a Cardiomyocyte Cell Line

**DOI:** 10.3390/biomedicines9070719

**Published:** 2021-06-23

**Authors:** Julia Hupfeld, Maximilian Ernst, Maria Knyrim, Stephanie Binas, Udo Kloeckner, Sindy Rabe, Katja Quarch, Danny Misiak, Matthew Fuszard, Claudia Grossmann, Michael Gekle, Barbara Schreier

**Affiliations:** 1Julius-Bernstein-Institute of Physiology, Medical Faculty of the Martin Luther University Halle-Wittenberg, 06112 Halle (Saale), Germany; julia_hupfeld@yahoo.de (J.H.); maximilian.ernst93@gmx.de (M.E.); maria.knyrim@medizin.uni-halle.de (M.K.); Stephanie.Binas@gmx.de (S.B.); udo.kloeckner@medizin.uni-halle.de (U.K.); sindy.rabe@medizin.uni-halle.de (S.R.); katja.quarch@medizin.uni-halle.de (K.Q.); claudia.grossmann@medizin.uni-halle.de (C.G.); michael.gekle@medizin.uni-halle.de (M.G.); 2Institute of Molecular Medicine, Medical Faculty of the Martin Luther University Halle-Wittenberg, Charles Tanford Protein Center, 06120 Halle (Saale), Germany; danny.misiak@medizin.uni-halle.de; 3Zentrum für Medizinische Grundlagenforschung, Core Facility—Proteomic Mass Spectrometry, Proteinzentrum Charles Tanford, Martin Luther University Halle-Wittenberg, 06120 Halle (Saale), Germany; matthew.fuszard@medizin.uni-halle.de

**Keywords:** miR-208b, cardiomyocytes, Kcnj5, Ran, Rock2, heart

## Abstract

MicroRNAs (miRs) contribute to different aspects of cardiovascular pathology, among them cardiac hypertrophy and atrial fibrillation. Cardiac miR expression was analyzed in a mouse model with structural and electrical remodeling. Next-generation sequencing revealed that miR-208b-3p was ~25-fold upregulated. Therefore, the aim of our study was to evaluate the impact of miR-208b on cardiac protein expression. First, an undirected approach comparing whole RNA sequencing data to miR-walk 2.0 miR-208b 3′-UTR targets revealed 58 potential targets of miR-208b being regulated. We were able to show that miR-208b mimics bind to the 3′ untranslated region (UTR) of voltage-gated calcium channel subunit alpha1 C and Kcnj5, two predicted targets of miR-208b. Additionally, we demonstrated that miR-208b mimics reduce GIRK1/4 channel-dependent thallium ion flux in HL-1 cells. In a second undirected approach we performed mass spectrometry to identify the potential targets of miR-208b. We identified 40 potential targets by comparison to miR-walk 2.0 3′-UTR, 5′-UTR and CDS targets. Among those targets, Rock2 and Ran were upregulated in Western blots of HL-1 cells by miR-208b mimics. In summary, miR-208b targets the mRNAs of proteins involved in the generation of cardiac excitation and propagation, as well as of proteins involved in RNA translocation (Ran) and cardiac hypertrophic response (Rock2).

## 1. Introduction

According to the World Health Organization, cardiovascular diseases are the leading cause of death worldwide, with functional and structural heart changes (“remodeling”) playing a major role. Cardiac remodeling is represented by changes in the size, shape and function of the heart due to cardiac injury or stress, often resulting in differential gene or protein expression [1]. Among the differentially expressed transcripts, several microRNAs (miRs) were observed [2,3]. miRs are conserved, non-coding RNAs with a size of 20–22 nucleotides regulating gene expression by posttranscriptional processes [4]. They bind mainly to the 3′ untranslated region (3′-UTR) of target mRNAs and thereby either repress translation or induce the degradation of mRNA [5]. The binding of miRNAs to the 5′-UTR, coding sequence and gene promoters has also been reported [6]. Interestingly, the binding of miRNAs to 5′-UTR or coding regions is supposed to have silencing effects, while binding to the promoter regions is supposed to induce transcription [7]. About 60% of human protein-coding genes harbor miRNA binding sites in their 3′-UTR [8]. By studying the interaction of miRNAs with 3′-UTRs of cardiac proteins, the impact of miRNAs on e.g., cardiac arrhythmia, myocardial infarction, valvular heart disease or heart development has been revealed (reviewed in e.g., [9,10]).

miRNAs show at least partially a tissue-specific expression pattern. miRNAs mainly expressed in the striated muscle are named myomiRs. Among them are miR-1, miR-133a, miR-206, miR-208a, miR-208b, miR-486 and miR-499 [11,12]. Most of these myomiRs are expressed in both the skeletal and the cardiac muscle. Exceptions are miR-208a, which is cardiac-specific, and miR-206, which is skeletal-muscle specific [13]. Among the myomiRs miR-208a, -208b and -499 share at least six nucleotides of the seed sequence [14,15]. The miR-208 family has been implicated, at least as biomarkers, in cardiac diseases [16,17], but most of the experiments were performed with regard to miR-208a, while miR-208b is considered mainly a “marker” miRNA [18,19]. miR-208a controls the expression of several genes that are transcriptional repressors of the β-MHC gene, indicating that these miRNAs might regulate muscle myosin content, myofiber identity and muscle performance [20]. Additionally, it has been demonstrated that the overexpression of miR-208a causes heart hypertrophy and changes in cardiac electrical conduction [14].

We analyzed miR expression in a genetic mouse model with severe cardiac hypertrophy [21] that is accompanied by electrophysiological changes [22]. Interestingly, the miR that was deregulated to the highest extent was miR-208b. To evaluate if miR-208b might have an impact on heart hypertrophy and cardiac electrical remodeling, we aimed to identify miR-208b targets. In two undirected approaches, we compared whole RNA sequencing data from hypertrophied mouse hearts, as well as mass spectrometry data from miR-208b-mimic-transfected mouse cardiomyocytes (HL-1 cells), to miR-walk 2.0 miR-208b targets.

Herein, we demonstrate that miR-208b regulates the expression of proteins involved in cardiac-excitation generation and propagation (Cacna1c, Kcnj5), as well as the proteins involved in RNA translocation (Ran) and cardiac hypertrophic response (ROCK2).

## 2. Materials and Methods

### 2.1. Animal Procedures

All mouse experiments described in this manuscript were approved by the local government (Landesverwaltungsamt Sachsen-Anhalt, Germany, permit number: 42502-2-1124 and -1201 MLU) and were performed according to the guidelines of the directive 2010/63/EU. All mice utilized in the experiments were of C57Bl6 background. Mice were kept in the facilities of the University of Halle-Wittenberg at a room temperature of 20 ± 1 °C and with a 12 h/12 h light/dark cycle. All adult animals were six months of age when included in the experiments. Generation, genotyping and the cardiovascular phenotype of EGFR KO animals were described before [21]. Briefly, these mice were generated by mating EGFR^flox/flox^ mice with SM22-Cre^tg^ mice, leading to a deletion of the EGFR in vascular smooth muscle cells and a strong reduction in cardiomyocytes, further on termed either EGFR^Δ/ΔVSMC&CM^ or knockout (KO).

For angiotensin II (AII, 1000 ng/kg BW/min over three weeks) or isoprenaline (iso, 30 mg/kg/day for two weeks) treatment, male animals were anaesthetized with isoflurane (~2% *v/v* in 100% O_2_, 1 L/min), and Alzet minipumps (1004) were implanted subcutaneously in the back of the animals. In each group, 5–10 animals were included in the study. Carprofen (5–10 mg/kg BW, Rimadyl, Pfizer, NY, USA) was injected subcutaneously immediately before pump implantation. If necessary, pain relief was repeated every 24 h for three days. Mice were sacrificed by cervical dislocation in isoflurane anesthesia. Hearts were removed, and weight was normalized to tibia length (HW/TL). Subsequently, the heart was divided for biochemical and histological analysis. Cardiomyocytes and cardiac fibroblasts were isolated from whole hearts as described before [23]. Fibroblasts were isolated through the incubation of the supernatant from the cardiomyocyte isolation overnight in petri dishes.

### 2.2. Gene Expression Analysis

For all analyses, total RNA was isolated either from whole hearts or isolated cells using the InviTrap spin tissue RNA mini kit (STRATEC, Berlin, Germany) or the TRIzol Reagent (Invitrogen, Darmstadt, Germany). One µg of total RNA was treated with DNase I (RNase-free) (NEB, Frankfurt, Germany), and reverse transcription (RT) was performed with random primers using SuperScript II reverse transcriptase (Invitrogen, Darmstadt, Germany), according to the manufacturer’s instructions.

Gene expression was analyzed via real-time qRT-PCR, and the mRNA amount was normalized to 18S rRNA or Gapdh. The sequence of primers, as well as annealing temperature and RefSeq accession number/id, are given in Appendix A.

To determine the absolute copy number of RNA, droplet digital PCR (ddPCR) was performed using the QX200 system of BioRad (Munich, Germany). cDNA was prepared as described above and used in ddPCR at the same conditions as in real-time qRT-PCR.

For TaqMan ddPCR, a primer pair and a FAM-labeled probe specific for miR-208b or miR-499 were used simultaneously with a primer pair and a HEX-labeled probe specific for U6 (Applied Biosystems, Karlsruhe, Germany). The list of TaqMan assays purchased from Applied Biosystems is given in Appendix A.

### 2.3. Next Generation Sequencing

#### 2.3.1. Small RNA Sequencing

Sequencing was performed as described previously [24]. RNA (500 ng) from each sample was used with the TruSeq Small RNA sample prep kit v2 (Illumina). The barcoded libraries were size-restricted between 140 and 165 base pairs (bp) for additional enrichment of miRs, purified and quantified using the Library Quantification Kit-Illumina/Universal (KAPA Biosystems, Woburn, USA). The sequencing of 50 bp was performed with an Illumina HighScan-SQ sequencer using version 3 chemistry and flow cell. All procedures were performed according to the instructions of the respective manufacturers. For the analysis of the differential expression of single miRNAs, we used the following thresholds: RPM ≥ 100 in WT for downregulated or ≥100 RPM in KO for upregulated miR, fold change ≥|1.5|, *p*-value < 0.01. For the prediction of miR-targets, mRNA enrichment analysis was performed by g:Profiler [25] and GOrilla [26]. The R packages DESeq2 and EdgeR were used for normalization and to calculate the differential expression of mRNAs, as reported previously [27].

#### 2.3.2. Total RNA Sequencing

Total RNA was isolated with TriZol Reagent, following the manufacturer’s instructions. rRNA was removed using RiboMinus (Life Technologies) according to the manufacturer’s instructions. RNA was fragmented (w/Tris-acetate, potassium acetate and magnesium acetate) and precipitated (w/100% ethanol, ammonium acetate and glycogen). Resulting RNAs were used to generate cDNA libraries using Illumina index adapters. Pools (measuring 10 nM) of up to 10 libraries were used for cluster generation (Illumina cBot). Total RNA-sequencing library preparation and sequencing was performed at the IKFZ (Leipzig, Germany). Low-quality read ends, as well as the remaining parts of sequencing adapters, were clipped using Cutadapt (v 1.6). Subsequently, reads were aligned to the mouse genome (UCSC GRCm38/mm10) using TopHat2 (v 2.0.13; [28]). FeatureCounts (v 1.4.6; [29]) was used for summarizing gene-mapped reads. Ensembl (GRCm38.77; [30]) was used for annotations. Differential expression was tested by a Poisson exact test [31] using TMM normalization, essentially as described previously [32]. Significant differential expression was determined by a significance level of 0.05 (FDR ≤ 0.05). mRNA enrichment analysis was performed by g:Profiler [25] and GOrilla [26].

### 2.4. Mass Spectrometry

HL-1 cells were transfected with miR-208b mimic or mimic control, as described for Western blot preparation. Cells were harvested with a cell-culture rubber and lysed in RIPA buffer supplemented with 1x Roche complete protease inhibitors. Mass spectrometric analyses were performed using the principles described earlier [33] with some alterations. Briefly, the lysate was submitted to three cycles of heat denaturation at 95 °C for 10 min and freezing at −80 °C for 10 min. DNA was sheared by sonication and afterwards, removed through the centrifugation of the lysate at 15,000× *g* for 30 min. Protein concentration was determined by BCA assay (Pierce, Thermo Scientific #23225) as per the manufacturer’s instructions. Isolated protein (50 µg), diluted in 10 µL modified RIPA buffer (with additional 0.1% RapiGest SF), was reduced with 20 mM DTT at 60 °C for 20 min, followed by cooling to room temperature for 10 min. The proteins were alkylated with 40 mM IAA for 45 min in darkness at room temperature. The alkylated proteins were filtered with passivated (5% Tween20-solution) Microcon Ultracel 30 kDa, and afterwards, three large immersions of ultrapure water were performed. Reduced and alkylated proteins were mixed with 80 µL 8 M urea and 0.2% DCA in 100 mM Tris-HCl pH 8 for detergent removal. The solution was filtered, and the result was then spun at 14,000× *g* for 15 min; the washing step was performed three times. Urea was removed using three washing steps with 50 mM ABC with 0.2% DCA by spinning at 14,000× *g* for 15 min. Afterwards, protein digestion was achieved by adding 1 μg of trypsin overnight at 37 °C. Peptides were recovered in two washes with 50 μL of 50 mM ABC while spinning at 14,000× *g* for 5 min. For peptide clean-up, C18 Spin Columns (Pierce, Thermo Scientific #89870) were utilized, as described in the manufacturer’s protocol. The eluted peptides were transferred to 1.5 mL Protein LoBind eppendorf tubes, then dried by SpeedVac and resuspended by adding 15 µL of 0.1% FA and sonicating for 10 min.

LC-MS/MS Measurements: Briefly, 1 µL of tryptic peptides (~400 ng peptides) was trapped on a 20 mm × 180 µm fused silica M-Class C18 trap column (Waters, Eschborn, Germany) and washed for 5 min at 5 µL/min with 0.1% formic acid (FA) in LC-MS grade water before being separated on a 250 mm × 75 µm fused silica M-Class HSS T3 C18 column with 1.8 µm particle size (Waters, Eschborn, Germany) over a 90 min gradient consisting of increasing concentrations of 7–35% ACN with 0.1% FA (Carl Roth, Karlsruhe, Germany). Eluting peptides were ionized at 2.1 kV from a pre-cut PicoTip Emitter (New Objective, Woburn, MA, USA) with source settings of 100 C and a nano N_2_ flow of 0.4 bars. Ions passed into the Synapt G2S Mass Spectrometer (Waters, Eschborn, Germany), which was operated in both the positive-ion mode and resolution mode with the following instrument settings: ion trap cell mobility separation with a release time of 500 μ. Afterwards, the result was “cooled” for 1000 μs; the helium pressure was set to 4.7 mbar, and the IMS cell nitrogen pressure was 2.87 mbar; the wave height was 38 V, and the wave velocity was ramped from 1200–400 m/s. Glu-1-Fibrinopeptide B (250 fmol/μL, 0.3 μL/min) was used as lock mass (m/z = 785.8426, z = 2). The Synapt was run in HDMSe mode acquiring over the range 50–2000 m/z for 1 s in low energy and 1 s in high energy. Peptides were separated in flight using Ion Mobility with settings as described above and fragmented at high energy with a Transfer CE ramp of 15–43 V.

The RAW MS mzML data files were generated with ProteinLynx Global Server (PLGS) 3.0.1 (Waters, Milford, MA, USA) with the following settings: automatic calculation of chromatographic peak width and MS TOF resolution; the lock mass for charge 2 was ‘785.8426 Da/e’; and thresholds were set to 135 counts for low energy, 80 counts for elevated energy and 750 counts for intensity, respectively. These mzML files were initially run through the sampling search engine Preview v3.3.11 (Protein Metrics, Cupertino, CA, USA) against *M. musculus* (UniProtKB proteome UP000000589, downloaded 08/2018), which generated full search parameters: a precursor mass tolerance of 20 ppm and a fragment mass tolerance of 30 ppm. Digestion specificity was set to trypsin with possible N-raggedness. The lock mass Glu-1-Fibrinopeptide B was used for the recalibration of fragments and precursors. A full PLGS database search against the above mouse proteome was set with these modifications: the fixed modification of carbamidomethylation on Cys, variable modifications of (di) oxidation on Met, N-terminal Gln->pyro-Glu and Glu->pyro-Glu conversion, N-terminal acetylation and the deamidation of Gln. The resulting outputs were subsequently loaded into the quantitative software IsoQuant [34] in order to evaluate experiment-wide protein abundances.

### 2.5. HL-1 Cell Line

HL-1 cells were maintained in Claycomb medium (Sigma, Munich, Germany) with the following supplements: 10% FCS (Biochrom, Berlin, Germany), 2 mM L-glutamine (Sigma), 100 µM noradrenaline (Sigma), 100 U/mL penicillin and 100 µg/mL streptomycin (Sigma).

HL-1 cells were transfected with 30 nM of miRCURY LNA miR-208b mimics or mimic negative control (Exiqon, Vedbaek, Denmark) using 5 µL Lipofectamine (Thermo Fisher Scientific, Waltham, MA, USA) in 1.5 mL DMEM (Biochrom, Berlin, Germany; without FCS) following manufacturer’s instructions. After 24 h, the medium was changed, and cells were kept on Claycomb medium with supplements for a further 48 h.

For Western blot analysis, HL-1 cells were washed with PBS and lysed in RIPA buffer and sonicated (UP100H, Hielscher, Teltow, Germany). Cell lysates were matched for protein content. After separation, the proteins were transferred to a nitrocellulose membrane (GE Healthcare, Buckinghamshire, UK) for protein detection. The membrane was incubated with primary antibodies (from Cell Signaling Technology, Danvers, MA, USA: ROCK2: 1:1000, 8236, Ran: 1:1000, 4462, β-actin (13E5): 1:1000, 4970, HSP90: 1:1000, 4874) at 4 °C overnight. The bound primary antibody was visualized using horseradish peroxidase-conjugated secondary IgG (anti-rabbit, 1:10,000 Rockland, Limerick, USA or for RAD18: anti-mouse: 1: 100,000, Rockland, Limerick, USA) and the ECLTM system (Amersham, Freiburg, Germany). Densitometry analysis was performed with Quantity One software (Biorad, Munich, Germany).

### 2.6. Electrophysiology

Single HL-1 cells were plated for 24 h on gelatin-/fibronectin-coated 35 mm Petri dishes in 2 mL of Claycomb medium with supplements. The cells were transfected with miR-208b mimics as described above. After transfection (48 h), measurements were performed in the whole-cell configuration of the patch-clamp technique using an Axopatch 200A patch-clamp amplifier (Axon Instruments, Inc., Burlingame, CA, USA) as described before [30]. Patch pipettes were fabricated from thick wall (2 mm OD) borosilicate glass capillaries (Hilgenberg, Malsfeld, Germany) and filled with an internal solution of the following composition (in mmol/L): 130 CsCl, 20 TEACl, 10 EGTA, 5 Na_2_ATP, 6 MgCl_2_ and 10 HEPES (pH was adjusted with CsOH to 7.2). The electrical resistance of the electrodes was 3–4 MΩ when filled with internal solution. L-type Ca^2+^ currents were recorded in a Na^+^-free and K^+^-free bath solution containing (in mmol/L): 150 Tris-Cl, 10 CaCl_2_, 10 glucose and 10 HEPES (pH was adjusted with Tris-OH to 7.4). Current signals were sampled at 16 to 40 kHz and low pass filtered at 5 kHz with a four-pole Bessel filter and stored for off-line analysis (ISO2, MFK, Germany). Series resistance was partially compensated (>70%). By integrating the capacitive current at the end of the 10 ms long voltage step (80 mV to −70 mV), the input capacitance of the cells was obtained. The peak amplitude of the inward current was normalized to the input capacitance to obtain the current density (pA/pF) in order to compensate for differences in cell size. All experiments were carried out at room temperature (20–24 °C). HL-1 cells express both T-type and L-type Ca^2+^ currents [35,36,37]. Because, in this study, we sought to investigate the effect of miR-208b solely on the activity of L-type Ca^2+^ channels, we used a voltage clamp protocol to separate the two currents from each other. T-type Ca^2+^ channels were inactivated by using a holding potential of −35 mV without affecting the availability of L-type Ca^2+^ channels. Initially, we performed current density measurements depolarizing the cells from a holding potential of −40 or −35 mV to various test potentials in 10 mV increments (from −40 mV to +65 mV). HL-1 cells were depolarized every 8 s for 100 ms from a holding potential of −35 mV to various test potentials (from +15 mV to 30 mV in 5 mV increments).

### 2.7. Dual Luciferase Reporter Assay

Dual luciferase reporter assay was described before [30]. Reporter constructs (pEZX-MT06 dual luciferase reporter) containing the murine 3′-UTRs of different mRNAs are listed in Appendix A. The vectors were transfected into HEK293 cells, 10 ng each. Additionally, the cells were transfected either with 30 nM of miRCURY LNA miR-208b mimics or mimic negative control using 1.5 µL Polyfect (Qiagen, Germantown, USA). After 24 h, the supernatant was removed and, after a further 48 h, cells were lysed, and the luciferase activity in the lysate measured using the Dual Luciferase Assay System (Promega, Madison, WI, USA). The firefly luciferase activity was normalized to the renilla luciferase activity. After that, values were normalized to mimic negative control as well as empty vector.

### 2.8. FluxOR Assay

HL-1 cells were transiently incubated with miR-208b mimics or mimic control for 24 h as described above. Some time (48 h) after the beginning of transfection, cells were seeded onto a 96 well plate and incubated for another 24 h. FluxOR II Green Potassium Ion Channel Assay (Invitrogen, Darmstadt, Germany) was performed according to manufacturer’s instructions using the Operetta CLS High-Content Analysis System (Perkin Elmer, Krakow, Poland) with a thallium ion (Tl^+^) concentration of 1 mM as described previously [30]. Fluorescence data were normalized to baseline fluorescence (F/F_0_). The time course of F/F_0_ was integrated to obtain the area under the curve (AUC). The carbachol effect (CE) was calculated as AUC (carbachol)-AUC (control) for each individual experiment and afterwards, normalized to the mean CE of the corresponding mimic control (scramble).

### 2.9. Statistical Analysis

Data are presented as mean ± standard error of mean (SEM). ANOVA, followed by post hoc testing, Student’s T-test or Mann–Whitney rank sum test were used, as applicable according to pre-test data analysis by Sigma Plot 12.5. A *p*-value < 0.05 was considered significant. Biometrical planning was performed with α = 0.05 and β = 0.8, resulting in sample sizes between 5 and 15 samples/group depending on the experimental setting. Graphics were prepared using Sigma Plot 12.5.

## 3. Results

### 3.1. miR-208b Overexpression Is Not Necessary for Heart Hypertrophy

As reported before, we analyzed miRNA expression in whole heart samples of mice with severe heart hypertrophy but without major signs of heart failure by next-generation sequencing [27]. Among the differentially expressed miRNAs, miR-208b-3p was the most prominent with an upregulation of ~25 fold in KO animals. First, we validated miR-208b upregulation via TaqMan qRT-PCR (Figure 1a) and ddPCR (Figure 1b) in a separate cohort of EGFR^Δ/ΔVSMC&CM^ mice. We could thereby demonstrate that, in whole heart samples, miR-208b is upregulated about 12-fold, confirming the findings from next-generation sequencing. In a former study, we demonstrated that, in our knockout model, the heart weight/tibia length ratio in male animals was significantly higher than in female animals [22]. Upregulation was not different between male (Appendix A) and female animals (Appendix A). Further on, we analyzed miR-208b expression and function regardless of gender. miR-208b has been proposed as a marker for heart hypertrophy [38]. To test the hypothesis that an increase in cardiac miR-208b indicates heart hypertrophy [39], we analyzed the expression of miR-208b and myh7 in hearts from mice treated with either angiotensin II or isoprenaline for three or two weeks, respectively. Both treatments resulted in an increase in heart weight to tibia length of ~25% [27]. In contrast to EGFR KO, in hearts from angiotensin II-treated or from isoprenaline-treated animals, there was no increase in miR-208b expression (Appendix A). We therefore concluded that increased miR-208b expression is not a prerequisite for heart hypertrophy. Detailed analysis of pri-miR-208b expression, host gene expression and the expression of the myomiRs 208a and 499 are given in the Appendix A. In summary, these data indicate that alterations in the expression of target genes from these three miRNAs in the native heart can only be attributed to alterations of miR-208b in cardiomyocytes.

### 3.2. mRNA Amount of the Predicted miR-208b Targets Cacna1c and Kcnj5 Was Reduced in Hearts of Mice with Heart Hypertrophy but Not in Mimic-Transfected HL-1 Cells

To identify possible target genes of miR-208b and thereby, generate a hypothesis about the function of miR-208b in the heart, we decided to perform an unbiased approach by reanalyzing a next-generation whole RNA-Seq dataset published before [27]. Employing miR-Walk 2.0, 3890 3′-UTRs were predicted by at least three out of 12 miR-databases as targets for miR-208b (Appendix A). We compared this list of genes to the differentially expressed genes in the hearts of mice with severe heart hypertrophy [27]. From 22,027 annotated protein coding RNAs, 968 had an abundance of at least 5 FPM in the wild-type animals, and Cohen D, a measure for the difference between the groups, was ≥2. Of these genes, 58 were predicted targets for miR-208b and differentially expressed between the two groups, with a downregulation of at least 1.5-fold in KO animals (Figure 2a, Appendix A). As expected, due to the number of regulated genes, gene-set enrichment analysis (GSEA) by G:Profiler showed no enrichment of genes with an adjusted *p*-value below 1.59 × 10^−3^. The lowest adjusted *p*-values could be obtained for the terms GTPase activator activity (*p* = 1.6 × 10^−3^), positive regulation of GTPase activity (*p* = 1.6 × 10^−3^) and valine, leucine and isoleucine (*p* = 1.9 × 10^−3^, Appendix A). If the fold change was increased to >2, only 14 genes met the criteria and only the term positive regulation of cardiac muscle contraction was determined to be enriched (*p* = 4.2 × 10^−2^). GOrilla revealed an enrichment of regulated genes involved in oxidoreductase activity (*p* = 3.3 × 10^−4^, FDR 1.1 × 10^−1^). An overview of the 58 regulated target genes of miR-208b is given in Appendix A. The results of the NGS analysis were validated by analyzing the expression of different target genes in an additional cohort of hearts. The genes were chosen according to the GSEA analysis and the cardiac electrical phenotype of the mice. As reported before [27], the transcription of Kcnj5, Cacnb2 and Cacna1c was reduced in the hearts of mice with severe heart hypertrophy. We were able to show that Mapk10, Ppm1k, RGS2, Adra1a and Slc25a22 were differentially expressed in KO compared to WT hearts but Kcnj2 was not (Figure 2b). SOD2 is a predicted target for miR-208b and is annotated in the GO-terms oxidation-reduction process, as well as oxidoreductase process. We analyzed the expression of this gene in the hearts of the animals via qRT-PCR but could not detect a differential mRNA-expression. To test if the differential expression in hearts is due to miR-208b, we transfected HL-1 cells with miR-208b mimics and analyzed the mRNA content after 48 h. Adra1a and Kcnj2 were not detectable in HL-1 cells after 48 h. miR-208b mimics did not reduce the mRNA amount of Mapk10, Slc25a22, SOD2, Kcnj5 or Cacnb2 in HL-1 cells compared to mimic control or non-transfected cells. Additionally, we analyzed if the validated target of miR-208b [40], Cacna1c, was downregulated by mimic transfection. In our hands there was no alteration of the transcription of this gene by miR-208b mimics (Figure 2c). From the selected target genes, only Ppm1k was slightly but significantly upregulated by miR-208b mimics (Figure 2c).

### 3.3. Fourty Potential miR-208b Targets Could Be Identified in HL-1 Cells by Mimic Transfection and Mass Spectrometry

In a second, undirected approach to identify potential miR-208b targets, HL-1 cells were transfected with mimics or mimic control, and mass spectrometry analysis on six samples from three independent cell passages per condition was performed. A total of 2566 proteins could be identified in at least one sample. Of these, 1558 could be detected in one of the groups in at least three out of six samples per group. Comparing mimic-control-transfected versus miR-208b-mimic-transfected HL-1 cells revealed that 120 proteins were at least 1.5-fold up- or downregulated. We compared the list of proteins identified by mass spectrometry to the miR-208b target genes predicted by miR-walk 2.0. At least 8375 proteins were predicted to have a miR-208b-binding site either in their 3′-UTR, the CDS or the 5′-UTR. Of the 120 differentially regulated proteins, 40 were predicted miR-208b target genes. Twenty-five were predicted due to binding sites in the 3′-UTR by at least three databases. Eleven proteins were predicted due to a binding site in the 5′-UTR and 15 due to binding sites in their CDS (Appendix A). Of these proteins, 18 were upregulated, and 22 were downregulated. GSEA analysis showed only a few GOterms or KEGG pathways to be significantly enriched, most probably due to the low amount of regulated proteins. Among these annotations the KEGG pathway “RNA transport” (adjusted *p*-value = 0.0006) and the GOterm “intracellular non-membrane-bounded organelle” (adjusted *p*-value = 0.006) were the ones with the lowest *p*-value. Among the annotated proteins, we chose Ran and Rock2 for further evaluation.

### 3.4. GIRK4-Dependent Ion Flux Is Reduced in HL-1 Cells by miR-208b Mimics

As the KO show electrical remodeling of the heart [22] and alterations in reactive-oxygen-species handling [21], we aimed to verify the interaction of miR-208b with the 3′-UTR of relevant target genes by dual luciferase assays in HEK-293 cells. At a concentration of 30 nM mimic or mimic control, there was no reduction in luciferase activity in the empty vector without the miR-208b binding site, while there was a significant reduction to 0.3- ± 0.06-fold (mean ± SEM, N = 4 experiments, *n* = 12 wells/group, Figure 3a) of the luciferase activity of the reporter plasmid carrying three miR-208b binding sites (positive control). As reported before [27], the 3′-UTR sequence of Cacna1c was divided into three fragments (Figure 3b). We were able to demonstrate that there was a slight but significant reduction in luciferase activity by miR-208b mimics in the 3′-UTR of Cacna1c II (Figure 3b). We could not detect an influence of miR-208b mimics on the 3′-UTRs of Cacnb2 (Figure 3b), Kcnd2 or SOD2 (Appendix A), but there was a small, significant reduction in the 3′-UTR luciferase activity of Kcnj5 (Figure 3b). To test if the binding of miR-208b to the 3′-UTR of Cacna1c II might lead to a functional reduction in L-type calcium-channel current, we transfected HL-1 cells with miR-208b mimic and mimic control and performed patch clamp analysis. In contrast to Canon et al. [40], we were not able to detect a significant difference in L-type calcium-channel current density (Figure 3c). To test if the binding of miR-208b might lead to a reduced ion current via GIRK1/4 we performed a fluxor assay. There was a significant reduction in carbachol-induced thallium current in miR-208b mimic-transfected HL-1 cells (Figure 3d).

In a next step, we aimed to validate the expression alteration of proteins detected by mass spectrometry. To reduce the number of proteins, we decided to (a) look for proteins where the variance was below 1% in the relative abundance in the two groups and (b) preferentially analyze proteins for which a technically adequate antibody was available. Therefore, we chose Rock2 and Ran. We could confirm by Western blot that Rock2 and Ran were both upregulated in HL-1 cells, as predicted by mass spectrometry (Appendix A).

## 4. Discussion

miRNAs are among the transcripts involved in cardiac remodeling [2,3]. MyomiRs have been implicated as markers for heart diseases [15,38,41,42,43], but the consequences of miRNA overexpression in the heart are still under investigation. Here we report on a mouse model with severe heart hypertrophy and electrical remodeling but without major signs of heart failure [21]. miR-208b was the most prominently upregulated miRNA in this model. It is a member of the miR-208 family of myomiRs sharing a very similar seed sequence. In two other models for heart hypertrophy, namely angiotensin II- and isoprenaline-induced heart hypertrophy, no increase in the expression of miR-208b could be observed. This indicates that the elevation of miR-208b is not necessary for the development of heart hypertrophy. Additionally, we were able to confirm in mice cubs, that, although their heart weight was already increased, miR-208b was not altered. This was comparable to the two clinically relevant heart hypertrophy models.

In the normal adult mouse heart, the main myosin heavy chain expressed is myh6/α-MHC. In contrast, in humans only about 10% of V1-MHC (αα) dimers exist [44], meaning that myh7/β-MHC is the main isoform [45]. Nonetheless, upon heart failure/heart hypertrophy, a switch in the expression pattern, leading to an increased/a more pronounced expression of myh6/β-MHC, occurs. Interestingly, miR-208a has been reported to enable that switch. miR-208a controls the expression of several genes that are transcriptional repressors of the β-MHC gene [20]. Together with studies in knockout and overexpression models [14], a contribution to structural remodeling can be assumed. miR-208a, miR-208b and miR-499 are intronic miRNAs of cardiac MHC genes [20]. It is believed that the expression of the host gene leads to the expression of the miRNA. In this manuscript, we demonstrate, that (i), while miR-208a is not altered in mouse cubs with heart hypertrophy, Myh6, the host gene of miR-208a, is reduced in these animals and (ii) that, while the miR-208b content in hearts from isoprenaline-treated animals is comparable to the untreated controls, the myh7 amount, the host gene miR-208b, is reduced. This might point towards a host-gene-independent, but regulated, expression of this miRNA. The mechanisms underlying the observed discrepancies have to be evaluated in a future study, especially as Monteys et al. [46] identified an intronic promoter upstream of miR-208b.

All members of the miR-208 family are differentially expressed in cardiac diseases [16,17] but to date mainly the effects of miR-208a have been evaluated. Grouping of miRNAs to families is founded by their seed sequence, but target recognition is not only due to these ~seven nucleotides [6]. First, we verified that miR-208a and miR-499 were not differentially expressed in our model for heart hypertrophy. To fulfill the task of identifying miR-208b targets in the heart, there are some difficulties to overcome. One of these difficulties is that the causes and consequences of heart hypertrophy are manifold, and additional adaptational processes occur during pathogenesis in vivo. To overcome these limitations, we employed two approaches to identify target genes for miR-208b. First, we analyzed the expression of miR-walk-predicted target genes in our in vivo mouse model, whereby we were able to identify Kcnj5 as a target gene for miR-208b. Interestingly, we already reported that Kcnj5 is a target of miR-221/222, the miR-cluster that increased the most in our model of heart hypertrophy [27]. Kcnj5 is the gene coding for the G protein-activated inward rectifier potassium channel 4 (GIRK4) and is one subunit of the I_K,ACh_ channel (GIRK1/4). Without this subunit, the I_K,ACh_ channel is not functional [47,48]. When acetylcholine binds to the muscarinergic M2-receptor with subsequent activation of the channel by G protein βγ subunits, the open probability of the channel increases [49]. Expression of I_K,ACh_-channels is higher in the atria than in the ventricles of several species [50,51], mimicking the physiological impact of the parasympathetic nervous system on the different cardiac structures. Functionally, the activation of I_K,ACh_ leads to the hyperpolarization of cardiomyocytes and a reduced action potential duration [52], thereby mediating the effect of the parasympathetic nervous system on heart rate and heart-rate variability [53]. Pathological parasympathetic input to the heart has been linked to different cardiac disorders, e.g., atrial fibrillation, atrioventricular block, heart failure and even sudden cardiac arrest (for review: [54]). Herein we demonstrate that miR-208b targets the Kcnj5 3´-UTR and reduces the thallium-ion current in HL-1 cells. Together with our findings that miR-221/222 reduce Kcnj5 current and L-type Ca^2+^-current in vitro, which are elevated in the same mouse model [27], this could be an interesting mechanism involved in cardiac arrhythmia [55,56]. Further studies will have to evaluate if the increased spontaneous mortality seen in these mice [21] is due to the alterations in miRNA expression.

In a second, undirected approach, we transfected HL-1 cells with miR-208b mimics and performed proteome analysis and further analyzed differentially expressed proteins. By this method, we identified Ran and ROCK2 as putative miR-208b targets. For these two proteins, future studies will take the form of analyzing the miRNA interaction with the mRNA.

In summary, we demonstrate that miR-208b as a marker for cardiac diseases regulates the expression of genes involved in excitation generation and propagation, namely Kcnj5 (GIRK4) and Cacna1c (calcium voltage-gated channel subunit alpha1 C), as well as RNA translocation (Ran) and cardiac hypertrophic response (ROCK2).

## Figures and Tables

**Figure 1 biomedicines-09-00719-f001:**
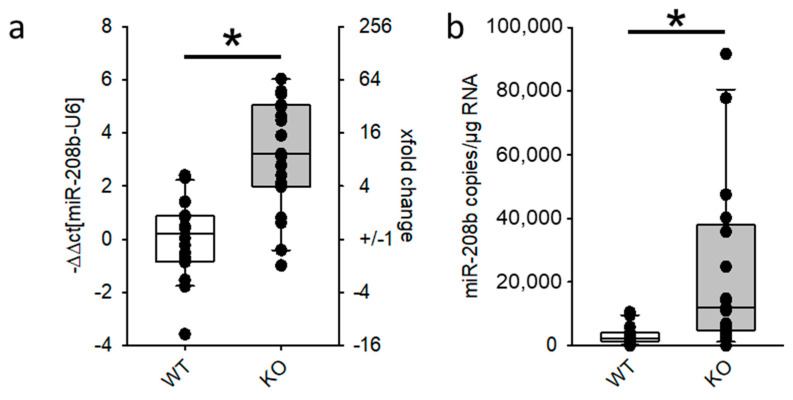
Expression of miR-208b in whole hearts from mice with severe heart hypertrophy. (**a**) Real-time qRT-PCR and (**b**) digital droplet PCR of male and female (N = 19–20 animals/group, * *p* ≤ 0.05 vs. WT).

**Figure 2 biomedicines-09-00719-f002:**
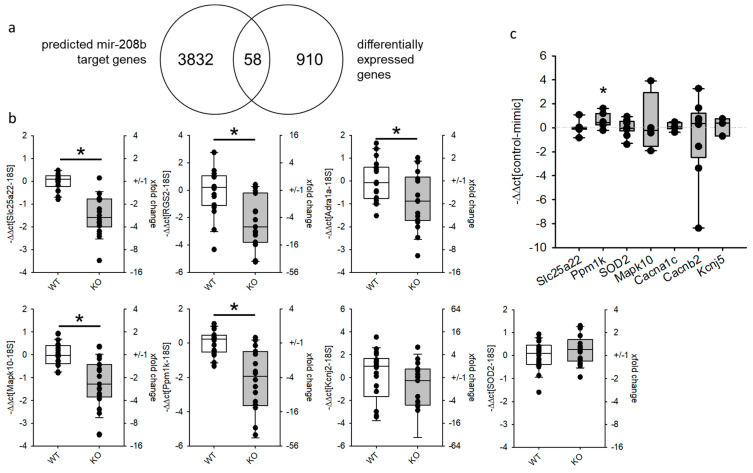
Differentially expressed target genes of miR-208b in the hearts of mice with severe heart hypertrophy. (**a**) Comparison of predicted miR-208b target genes (3′-UTR, miR-walk 2.0) with downregulated genes identified by next-generation whole RNA sequencing (whole-heart WT versus KO, N = 6 animals/group) that revealed 58 common genes. (**b**) Real-time qRT-PCR for selected genes in additional whole-heart samples of mice with severe heart hypertrophy (N = 15–19 animals/group, * *p* ≤ 0.05 vs. WT). (**c**) Real-time qRT-PCR for selected miR-208b target genes in HL-1 cells transfected either with miR-208b mimics or mimic control (N = 3–7 samples/group, * *p* ≤ 0.05 vs. mimic control).

**Figure 3 biomedicines-09-00719-f003:**
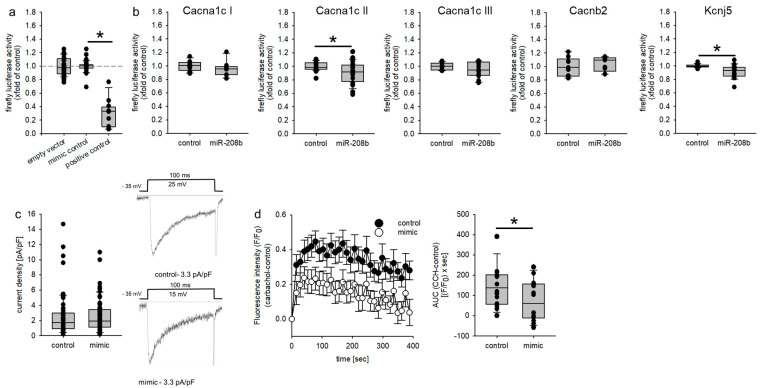
(**a**) HEK293 cells were transfected with either an empty vector or a luciferase vector containing a miR-208b-binding site. Co-transfection with mimic control had no significant effect on firefly luciferase activity, while co-transfection with the mir-208b mimic reduced firefly luciferase activity significantly (N = 3–12 experiments/group, n = 12–36 wells/group, * *p* ≤ 0.05 vs. mimic control). (**b**) To evaluate if the miRNA binds to the 3′-UTR dual luciferase constructs containing the 3′-UTR from the L-type Ca2+ channel subunits (Cacna1c, Cacnb2) or the GIRK4 potassium channel subunit (Kcnj5), the seed sequence or an empty vector were transfected in HEK293 cells either with or without mimic control or miR-208b mimic. miR-208b mimic reduced the luciferase activity of the Cacna1c-II and the Kcnj5 construct (N = 3–8 experiments/group, n = 9–24 wells/group, * *p* ≤ 0.05 vs. mimic control). (**c**) To analyze the effect of miR-208b on the L-type Ca^2+^ channel, we performed patch clamp analysis in HL-1 cells transfected with mimic control or miR-208b mimics. miR-208b had no effect on current density in HL-1 cells (N = 7 experiments, n = 55–60 cells/group). Representative current tracings for control and mimic are given. (**d**) HL-1 cells were transfected either with mimic control or miR-208b mimics, and GIRK4-dependent ion flux was measured through the fluorescence changes of a thallium-sensitive dye. In HL-1 cells transfected with miR-208b mimics, the area under the curve and thereby, the ion flux over time were significantly reduced compared to control cells. (N = 5 experiments, n = 15 wells, * *p* ≤ 0.05 vs. mimic control).

## Data Availability

The data presented in this study are available on request from the corresponding author.

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
