# Peer review of "miR-208b Reduces the Expression of Kcnj5 in a Cardiomyocyte Cell Line"

_biomedicines, 2021, doi:10.3390/biomedicines9070719_

Round 1

Reviewer 1 Report

file attached

Reviewer 2 Report

The authors have provided an interesting story with studying how miR-208b is involved with cardiac disease. They have provided a clear, concise, and well written paper on their findings. I believe that after addressing a few concerns I have this manuscript will have value and interest to the scientific community.

Major Concerns:

  1. The authors performed both small RNA-seq and total RNA-seq for this study. Although miR-208b was prominently upregulated in the small RNA-seq, they neglect to show any other data related to this. The authors also need to show the overall global miRNA-seq profile comparing KO mice to WT mice. I would imagine this will also bring other miRNAs of interest related to cardiac impact. I would imagine the authors would then be able to define an overall miRNA signature that will be related to what they observed. In addition, for both the small RNA-seq and total RNA-seq data additional pathway analysis can be done to show the overall regulation of both the downstream targets of the miRNAs and the genes themselves. This then can be related to the mass spec data which they have presented. Again the analysis done with the mass spec, is lacking pathway analysis or any detailed analysis that can assist with strengthening the results and determine what key pathways are being regulated with the miRNAs. Overall, the bioinformatic analysis in this paper is lacking and should be added to the paper.
  2. For all box plots in the figures, the authors should also show individual data points that created the box plot. This will give the reader more confidence with the data distribution and also will clearly show the number of biological replicates utilized for each comparison (for qPCR, you should not show or use the technical replicates with the analysis and only utilize biological replicates).
  3. Figure 7: To the eye in the westerns, the overexpression of Ran and ROCK2 is rather hard to identify. For the box plots quantifying this, the authors should also show the individual data points as I discussed above.

Minor Concerns:

  1. Methods, Pg, 2 Animal Procedures: The authors should state what the background for the EGFR KO mice are. That way a reader will not have to look up the reference they have provided.
  2. Results: pg.9: Remove the addition "in" in the first sentence of the second paragraph.

Reviewer 3 Report

The study conducted by Hupfeld, et al analyzed miR expression in EGFR KO model and found miR-208b had the highest extend. Later experiments demonstrated that miR-208b targeted proteins involved in cardiac excitation generation and cardiac hypertrophic response.

Here are some concerns that need to be addressed:

  1. even though the mouse model was described in a previously published paper, it is still necessary to have a more detailed description of it. Is this model a conditional transgenic model? what cre was used?
  2. Please define the round dots shown in all box plot figures. Are those individual data points or outliers?
  3. What is the age of mice used in figureS3C? why is the difference of myh6 between female and male alters in FIgure 3g?

Round 2

Reviewer 1 Report

The manuscript entitled “miR-208b reduces the expression of Kcnj5, Rock2 and Ran in a cardiomyocyte cell line” includes a large amount of experimental data with diverse cutting-edge technologies. Individual experiments were carefully done, and the paper is mostly well written. Yet, the overall structure of this paper is very awkward: the paper title, its purpose and collection of experimental data seem mismatched. It looks like a paper that includes everything that the authors have done without a hypothesis to be tested or overall purpose.
Major issue
The paper starts with the identification of miRNAs whose expression is altered in egfr-ko mouse hearts. The authors then try to figure out what roles of the identified miR-208b may have on the heart functions. It appears that miR-208b has nothing to do with the induction of cardiac hypertrophy, which seems the only change in egfr-ko mouse hearts in the authors’ mind. Ordinarily this seems the end of testing the hypothesis that the authors could have had. The paper, however, goes on to look for potential targets of this microRNA in the heart of this genetically modified and other animals, as well as in a cardiomyocyte-like cell line. The experiments using transfection of miRNA mimic revealed that the expression and function of Kcnj5 (GIRK4) is affected by miR-208b in this cell line. A reasonable connection of the front part to the latter portion could be made, if GIRK current reduces in cardiomyocytes of egfr-ko mice, but not in those in angiotensin II- or isoproterenol-induced hypertrophied myocytes. Otherwise, various experiments and data would make readers just too confused.
This reviewer recommends that the authors should just focus on one clear finding of miR-208b effects in a cell line without a large front portion, although biological or pathological significances of the finding remains to be seen. Indeed, this is what the title of the paper indicates.
Minor issues
1. Subtitles of the result section sound experimental procedures, such as mRNA expression, protein expression and functional evaluation. This is not a proper way to organize the manuscript. Subtitles should be a concise statement of major findings. Accordingly, redundant explanation for mRNA, protein or function should be reduced.
2. When starting with miRNA, M should not be a large capital.
3. Alignment is centered in some parts of the paper
4. Number+fold needs a space or hyphen in between.
5. I am not sure why the authors used indirect measurement for potassium current. Instead, patch clamp recording (used for calcium currents in the paper) is more direct and better evaluation of GIRK current.
6. I feel no compelling reasons to measure Ran and ROCK protein levels. Moreover, I would hesitate to include their changes in the title, as their changes seem small.

Author Response

Reviewer: The manuscript entitled “miR-208b reduces the expression of Kcnj5, Rock2 and Ran in a cardiomyocyte cell line” includes a large amount of experimental data with diverse cutting-edge technologies. Individual experiments were carefully done, and the paper is mostly well written.

Yet, the overall structure of this paper is very awkward: the paper title, its purpose and collection of experimental data seem mismatched. It looks like a paper that includes everything that the authors have done without a hypothesis to be tested or overall purpose.

Authors reply: We tried to work out the details of our hypothesis:

Page 2, last paragraph of the introduction: “. To evaluate if miR-208b might have an impact on heart hypertrophy and cardiac electrical remodeling, we aimed to identify miR-208b targets. In a first undirected approach we compared whole RNA sequencing data from hypertrophied mouse hearts to miR-walk 2.0 miR-208b targets. Subsequently, in a second, undirected approach to identify miR-208b targets we performed mass spectrometry with HL-1 cells transfected with miR-208b mimics or mimic control.”

Page 7, first paragraph: “Interestingly, in a former study we demonstrated that in our knockout model the heart weight/ tibia length ratio in male animals was significantly higher than in female animals [25]. To test the hypothesis that an increase in cardiac miR-208b indicates a heart hypertrophy, we analyzed the expression of miR-208b and myh7 in hearts from mice treated either with angiotensin II or isoprenaline for three or two weeks, respectively. Both treatments resulted in an increase of heart weight to tibia length of ~25% [30]. In contrast to EGFR KO, in hearts from angiotensin II-treated or from isoprenaline-treated animals there was no increase in miR-208b expression (Supplementary figure 1A, B). We therefore concluded that increased miR-208b expression is not a necessary prerequisite for heart hypertrophy.”

Page 9, first paragraph: “miR-208b is embedded in the myh7 (β-MHC) gene [16] and part of the miR-208 myomiR family. Therefore, we sought to get an overview about the expression of the miR-208 family members and their respective host genes.”

Page 11, first paragraph: “To identify possible target genes of miR-208b and thereby generate a hypothesis about the function of miR-208b in the heart, we took advantage of a next generation whole RNA-Seq data set published before [30].”

Reviewer: The paper starts with the identification of miRNAs whose expression is altered in egfr-ko mouse hearts. The authors then try to figure out what roles of the identified miR-208b may have on the heart functions. It appears that miR-208b has nothing to do with the induction of cardiac hypertrophy, which seems the only change in egfr-ko mouse hearts in the authors’ mind.

Authors reply: The EGFR-KO mice show a very prominent heart hypertrophy which is accompanied with electrical remodeling but only with minor signs of fibrosis or heart failure. As the electrical remodeling found in the mice was one of the reasons why we preferentially investigated the ion channels, we now tried to make this thoughts more comprehensible by adding the following to the manuscript:

Page 2, last paragraph of the introduction: “To evaluate if miR-208b might have an impact on heart hypertrophy and cardiac electrical remodeling, we aimed to identify miR-208b targets.”

Page 11, first paragraph: “To validate the results of the NGS analysis we analyzed the expression of different target genes in an additional cohort of hearts according to the GSEA analysis and the cardiac electrical phenotype of the mice.”

Reviewer: Ordinarily this seems the end of testing the hypothesis that the authors could have had. The paper, however, goes on to look for potential targets of this microRNA in the heart of this genetically modified and other animals, as well as in a cardiomyocyte-like cell line.

The experiments using transfection of miRNA mimic revealed that the expression and function of Kcnj5 (GIRK4) is affected by miR-208b in this cell line.

A reasonable connection of the front part to the latter portion could be made, if GIRK current reduces in cardiomyocytes of egfr-ko mice, but not in those in angiotensin II- or isoproterenol-induced hypertrophied myocytes. Otherwise, various experiments and data would make readers just too confused.

This reviewer recommends that the authors should just focus on one clear finding of miR-208b effects in a cell line without a large front portion, although biological or pathological significances of the finding remains to be seen.  Indeed, this is what the title of the paper indicates.

Authors reply: Although we see the point the reviewer has, we would like to include the front part of the paper for different reasons. First, we think that the results summarized in this part are of scientific value and should be reported. Second, the results explain why conclusions need to be drawn carefully from our experiments and why further investigations on miR-208b function in different models of heart hypertrophy need to be performed. And third, the other reviewers did not recommend to exclude these data from this manuscript, which brings us to the difficult situation that we need to disappoint at least one of the reviewers. 

Reviewer: Subtitles of the result section sound experimental procedures, such as mRNA expression, protein expression and functional evaluation. This is not a proper way to organize the manuscript.  Subtitles should be a concise statement of major findings.  Accordingly, redundant explanation for mRNA, protein or function should be reduced. 

Authors reply: We changed the subtitels in the results section accordingly:

3.1. miR-208b overexpression is not necessary for heart hypertrophy.

3.2. mRNA amount of the predicted miR-208b targets Cacna1c and Kcnj5 was reduced in hearts of mice with heart hypertrophy, but not in mimic transfected HL-1 cells.

3.3. Fourty potential miR-208b targets could be identified in HL-1 cells by mimic transfection and mass spectrometry.

3.4. GIRK 4-dependent ion flux is reduced in HL-1 cells by miR-208b mimics.

To reduce the redundant explanation for mRNA, protein or function, we deleted the following parts of the manuscript:

Page 12, last paragraph, first sentence: miRNAs might alter protein expression by reducing translation without affecting mRNA abundance

Reviewer: When starting with miRNA, M should not be a large capital.

Authors reply: We reread the manuscript thoroughly and replaced every large capital M in miRNA.

Reviewer: Alignment is centered in some parts of the paper

Authors reply: We corrected the formatting of the text.

Reviewer: Number+fold needs a space or hyphen in between.  

Authors reply: We added a space between a number and the word fold.

Reviewer: I am not sure why the authors used indirect measurement for potassium current.  Instead, patch clamp recording (used for calcium currents in the paper) is more direct and better evaluation of GIRK current.

Authors reply: We agree with the reviewer that measuring potassium current via patch clamp would be beneficial in comparison to measuring the thallium current indirectly using fluorescence microscopy. We performed patch clamp recordings for the GIRK current, but unfortunately only 10-20 percent of HL-1 cells do express the GIRK 1/4 channels. Therefore, we decided to use an approach enabling us to measure the fluorescence signal of several cells at a time on a low baseline background, namely intracellular thallium concentration change instead of intracellular potassium concentration change.

Reviewer: I feel no compelling reasons to measure Ran and ROCK protein levels. Moreover, I would hesitate to include their changes in the title, as their changes seem small.

Authors reply: We excluded Ran and Rock2 from the titel.

Reviewer 3 Report

concerns were well addressed

Author Response

Thank you very much for your efforts.

Round 3

Reviewer 1 Report

file attached

Author Response

Reviewer:

The authors revised the manuscript by eliminating, adding and changing some sentences. Yet, the overall revision seems very superficial.  I still feel a large gap between the title and the content.  It is still very hard to follow the entire paper.  In addition, the revision seems rough without careful inspections.

As the title of the paper indicates, the major finding in this study is that miR-208b reduces the expression of Kcnj5 in cardiomyocyte-like cells.  If I remember correctly, there are many articles about Kcnj5 and cardiac arrythmias, such as atrial fibrillation.  Moreover, the authors’ previous study suggests that egfr-ko mice are more likely to develop cardiac arrythmias with aldosterone administration and high salt intake.  Again, many articles report the connection among Kcnj5, aldosterone and atrial fibrillation.  Given this large volume of knowledge, I would think that the authors could make more logical flow in the entire article from introduction to discussion.

Authors reply:

We rearranged the manuscript and focused it on Kcnj5 and its function in the heart. There is now a detailed part in the discussion regarding this topic. Parts that are not necessary were shortened or removed. In detail we changed the following (only the current text is given due to the extensive changes):

Abstract, line 3: severe heart hypertrophy was removed, “structural and electrical remodeling” was inserted.

Introduction, page 2, second paragraph:

Among the myomiRs miR-208a, -208b and -499 share at least six nucleotides of the seed sequence [14,15]. The miR-208 family has been implicated, at least as biomarkers, in cardiac diseases [16,17], but most of the experiments were performed with regard to miR-208a, while miR-208b is considered mainly a “marker” miRNA [18,19]. miR-208a controls the expression of several genes that are transcriptional repressors of the β-MHC gene, indicating that these miRNAs might regulate muscle myosin content, myofiber identity and muscle performance [20]. Additionally, it has been demonstrated, that overexpression of miR-208a causes heart hypertrophy and changes in cardiac electrical conduction [14].

Introduction, page 2, third paragraph:

Interestingly, the miR that was deregulated with the highest extend was miR-208b. To evaluate if miR-208b might have an impact on heart hypertrophy and cardiac electrical remodeling, we aimed to identify miR-208b targets.

Results, page 7, first paragraph:

miR-208b has been proposed as a marker for heart hypertrophy [38]. In a former study we demonstrated that in our knockout model the heart weight/tibia length ratio in male animals was significantly higher than in female animals [22]. To test the hypothesis that an increase in cardiac miR-208b indicates a heart hypertrophy [39], we analyzed the expression of miR-208b and myh7 in hearts from mice treated with either angiotensin II or isoprenaline for three or two weeks, respectively. Both treatments resulted in an increase of heart weight to tibia length of ~25% [27]. In contrast to EGFR KO, in hearts from angiotensin II-treated or from isoprenaline-treated animals there was no increase in miR-208b expression (Supplementary figure 2 A, B). We therefore concluded that increased miR-208b expression is not a prerequisite for heart hypertrophy. Detailed analysis of pri-miR-208b expression, host gene expression and expression of the myomiRs 208a and 499 are given in the supplementary material (supplementary figure S2-S6). In summary, these data indicate that alterations in the expression of target genes from these three miRNAs in the native heart can only be attributed to alterations of miR-208b in cardiomyocytes.

Discussion, page 15, paragraph 1:

Here we report on a mouse model with severe heart hypertrophy, electrical remodeling but without major signs of heart failure [21][24].

Discussion, page 15, paragraph 1&2:

This was comparable to the two clinically relevant heart hypertrophy models.

In the normal adult mouse heart, the main myosin heavy chain expressed is myh6/α-MHC.

Discussion, page 15, paragraph 2:

miR-208a controls the expression of several genes that are transcriptional repressors of the β-MHC gene [20] together with studies in knockout and overexpression models [14] a contribution to structural remodeling can be assumed.

.... This might point towards a host-gene independent but regulated expression of this miRNA. The mechanisms underlying the observed discrepancies have to be evaluated in a future study, especially as Monteys at al. [46] identified an intronic promoter upstream of miR-208b.

Discussion, page 16, paragraph 1:

To overcome these limitations we employed two approaches to identify target genes for miR-208b.

.... Without this subunit the IK,Ach channel is not functional [47,48]. When acetylcholine binds to the muscarinergic M2-receptor with subsequent activation of the channel by G protein βγ subunits the open probability of the channel increases [49]. Expression of IK,Ach-channels is higher in atria than in ventricles of several species [50,51], mimicking the physiological impact of the parasympathetic nervous system on the different cardiac structures. Functionally, activation of IK,Ach leads to hyperpolarization of cardiomycytes and a reduced action potential duration [52]. Thereby mediating the effect of the parasympathetic nervous system on heart rate and heart rate variability [53]. Pathological parasympathetic input to the heart has been linked to different cardiac disorders, e.g. atrial fibrillation, atrioventricular block, heart failure and even sudden cardiac arrest (for review: [54]). Herein we demonstrate that miR-208b targets the Kcnj5 3´-UTR and reduces thallium ion current in HL-1 cells. Together with our findings that miR-221/222 reduce Kcnj5 current and L-type Ca2+-current in vitro and are elevated in the same mouse model [27], this could be an interesting 

mechanism involved in cardiac arrhythmia [55,56]. Further studies will have to evaluate, if the increased spontaneous mortality seen in these mice [21] is due to the alterations in miRNA expression.

Discussion, page 16-17, paragraph 2-4:

In a second, undirected approach we transfected HL-1 cells with miR-208b mimics and performed proteome analysis and further analyzed differentially expressed proteins. By this, we identified Ran and ROCK2 as putative miR-208b targets. For these two proteins future studies will take the form of analyzing the miRNA interaction with the mRNA.

In summary, we demonstrate that miR-208b as a marker for cardiac diseases regulates the expression of genes involved in excitation generation and propagation, namely Kcnj5 (GIRK4) and Cacna1c (calcium voltage-gated channel subunit alpha1 C) as well as RNA translocation (Ran) and cardiac hypertrophic response (ROCK2).

Reviewer:

The authors wish to include a large volume of data that are not directly related to the main finding of the paper, because these data could be useful for someone in future.  If so, I recommend the authors to move those data unnecessary to make a logical connection to the supplemental section.

Authors reply:

We moved figure 1 b,c, e, f, 2, 3, 5 and 7 to the supplements.